# Structure and predictors of in-hospital nursing care leading to reduction in early readmission among patients with schizophrenia in Japan: A cross-sectional study

**Shigeyoshi Maki**[1,¤,*], **Kuniyoshi Nagai**[2], **Shoko Ando**[3], **Koji Tamakoshi**[3]

1 Department of Nursing, School of Nursing, Sugiyama Jogakuen University, Nagoya, Aichi, Japan,
2 Department of Nursing, School of Nursing, Nagoya University of Arts and Sciences, Nagoya, Aichi, Japan,
3 Department of Nursing, Nagoya University Graduate School of Medicine (Health Sciences), Nagoya, Aichi, Japan

☯ These authors contributed equally to this work.
¤ Current address: Nursing Course, School of Medicine, Gifu University, Gifu, Japan
\* s-maki@sugiyama-u.ac.jp

**Data Availability Statement:** All relevant data are within the manuscript and its Supporting Information files.

## Abstract

Schizophrenia is a disorder characterized by psychotic relapses. Globally, about 15%–30% of patients with schizophrenia discharged from inpatient psychiatric admissions are readmitted within 90 days due to exacerbation of symptoms that leads to self-harm, harm to others, or self-neglect. The purpose of this study was to investigate the structure and predictors of in-hospital nursing care leading to reduction in early readmission among patients with schizophrenia. A new questionnaire was developed to assess the extent to which respondents delivered in-hospital nursing care leading to reduction in early readmission among patients with schizophrenia. This study adopted a cross-sectional research design. The survey was conducted with the new questionnaires. The participants were registered nurses working in psychiatric wards. Item analyses and exploratory factor analyses were performed using the new questionnaires to investigate the structure of in-hospital nursing care leading to reduction in early readmission. Stepwise regression analyses were conducted to examine the factors predicting in-hospital nursing care leading to reduction in early readmission. Data were collected from 724 registered nurses in Japan. In-hospital nursing care leading to reduction in early readmission was found to consist of five factors: promoting cognitive functioning and self-care, identifying reasons for readmission, establishing cooperative systems within the community, sharing goals about community life, and creating restful spaces. In-hospital nursing care leading to reduction in early readmission was predicted by the following variables: the score on the nursing excellence scale in clinical practice, the score on therapeutic hold, and the participation of community care providers in pre-discharge conferences. Japanese psychiatric nurses provide nursing care based on these five factors leading to reduction in early readmission. Such nursing care would be facilitated by not only

**Funding:** SM was funded by Japan Society for the Promotion of Science KAKENHI Grant# JP17K17523. URL: https://kaken.nii.ac.jp/grant/KAKENHI-PROJECT-17K17523/ The funders had no role in study design, data collection and analysis, decision to publish, or preparation of the manuscript.

**Competing interests:** The authors have declared that no competing interests exist.

nurses' excellence but also nurses' environmental factors, especially the therapeutic climate of the ward and the participation of community care providers in pre-discharge conferences.

## Introduction

Schizophrenia is a disorder characterized by psychotic relapses [1]. Many people with schizophrenia suffer recurrent symptoms of psychosis (i.e., hallucinations, delusions, and disorganized speech) and chronic cognitive deficits (i.e., impaired executive function, memory, and speed of mental processing) [2]. Various indicators are used for the relapse of schizophrenia. Readmission rate is widely used as an indicator of relapse of schizophrenia [3, 4]. Some patients with schizophrenia have been readmitted to hospitals because their condition worsens so that they cannot live in communities [2]. Readmissions lead to rising costs of mental health care [3–5]. Furthermore, readmissions put a strain on patients, affect their prognosis, and lead to a breakdown of their family function [3, 6].

Several studies examined the factors that contribute to readmission. Schizophrenia is one of the diagnoses of mental disorders associated with psychiatric readmission [7–9]. Factors associated with readmission can be classified based on three criteria: patients' clinical characteristics, health system characteristics, and characteristics of hospitalizations. Regarding patients' clinical characteristics, younger age [10, 11], marital status of "unmarried" [11], complications [6, 12], medication nonadherence [13–15], and maladaptive functioning of family systems [16] are associated with increased risk for readmission. Regarding health system characteristics, proportion of experienced psychiatrists at a hospital [17], multiple uses of health service [6], and unplanned discharge [12] are associated with increased risk for readmission. Regarding characteristics of hospitalizations, having a history of previous hospitalizations [6, 11, 12], duration of involuntary admission [11, 12], and total admission duration [6, 11] are associated with increased risk for readmission. Furthermore, a length of stay in a psychiatric ward of more than 28 days plays a protective role in preventing readmission [4].

In particular, readmissions within 90 days of discharge are usually defined as early readmissions [10]. Between 15%–30% of patients with psychiatric disorders experience early readmission [4, 11, 18, 19]. Early readmissions reflect the degree of continuity between hospital care and community care [10]. Patients take an average of 90 days to establish therapeutic relationships with community care providers after their discharge [20]. Many patients encounter critical situations within 90 days of discharge [21, 22]. Reduction in early readmission could lead to successful transitions of patients from hospital care to community care. In-hospital care that leads to reduction in early readmission is important for achieving successful transitions from hospital care to community care.

Revealing the structure of such in-hospital care will facilitate the development of intervention programs that support this transition. Previous studies have identified the structure of the following six types of in-hospital psychiatric care: care and management of inpatients who exhibit self-cutting behaviors [23]; case management practice [24]; health services to promote mental health [25, 26]; psychiatric services in forensic inpatient care [27]; therapeutic attitudes of professionals working with drug abusers [28]; and rehabilitative care in longer term mental health facilities [29–32]. However, the structure of in-hospital care leading to reduction in early readmission is unclear.

Most psychiatric care in Japan is still provided in inpatient settings [33], despite Japan's efforts to facilitate the transition from inpatient care to community care [34]. The number of

psychiatric beds in Japan is four times higher than the Organization for Economic Cooperation and Development (OECD) average (267 beds vs. 66 beds per 100,000 population) [35]. The average length of stay in a psychiatric hospital in Japan is approximately 300 days, much longer than in other OECD countries [36, 37]. Community mental health services in Japan are not sufficient [37]. Approximately 25% of inpatients experience readmissions within 90 days of discharge (i.e., early readmissions) [38]. In Japan, delivering community mental health services for patients in communities are not enough to decrease early readmission in patients [4]. The context of mental health care in Japan is different from elsewhere in the world in that the length of hospitalization is longer and the more resources are invested in inpatient care rather than community care.

Therefore, it is desirable to identify the predictors of in-hospital nursing care leading to reduction in early readmission in the Japanese mental health care context. Specifically, this information could be used to develop strategies to support the successful transition of patients from hospital care to community care. In-hospital nursing care leading to reduction in early readmission may be predicted by nurses' environmental factors as well as their individual factors. The risk of early readmission could be predicted by not only community follow-ups but also by length of stay in hospital [4]. Nursing performance is influenced by nurses' environmental factors as well as personal factors [39]. A positive social climate in hospitals is associated with higher patient treatment motivation, treatment engagement, and patient-nurse therapeutic alliance [40]. A significant relationship between social climate and staff performance and morale has been reported [41]. We hypothesized that in-hospital nursing care leading to reduction in early readmission could also be predicted by the social climate in psychiatric wards as well as by nurses' individual factors.

The purpose of this study was: (a) to investigate the structure that underlies in-hospital nursing care leading to reduction in early readmission among patients with schizophrenia; and (b) to examine the factors predicting in-hospital nursing care leading to reduction in early readmission, by focusing on both the social climate in psychiatric wards and nurses' individual factors.

## Methods

### Design

This study adopted a cross-sectional research design and utilized self-administered questionnaires, which participants completed anonymously.

### Settings and participants

This survey was conducted at hospitals where psychiatric beds accounted for more than 60% of all beds. The hospitals were selected in three steps. First, the hospitals were extracted from the Japan Medical Analysis Platform of the Japan Medical Association (http://jmap.jp/). Of the 1,271 hospitals extracted, 59 were public hospitals, 69 were non-profit corporation hospitals, and 1,143 were private hospitals. Second, the 128 public hospitals and hospitals owned by non-profit corporations were all included in the study, as well as a randomly selected sample of 128 private hospitals, which totaled 256 hospitals. Third, we then mailed request documents for research cooperation to the directors of the 256 hospitals. Written consents for research cooperation were obtained from 40 hospitals (i.e., 19 public hospitals, six hospitals owned by non-profit corporations, and 15 private hospitals).

Nurse managers at the 40 selected hospitals provided information on the number of registered nurses (RNs) working at their hospital who met the inclusion criteria listed below. They reported that 1,995 RNs fulfilled the criteria. Subsequently, request documents for research

cooperation to the RNs with self-administered questionnaires were mailed to the nurse managers, who then distributed them to the 1,995 RNs identified. The RNs, who gave written consent for research cooperation, returned the completed questionnaires to the researchers by mail. The inclusion criteria were: 1) full-time RN, and 2) working in a psychiatric ward. The exclusion criteria were: 1) nurse managers, 2) RNs working in outpatient wards, and 3) part-time nurses. Data were collected between February and March 2018.

Sample size estimations were based on the recommendations of Pett, Lackey & Sullivan [42]. They recommended that the minimum number of subjects for an exploratory factor analysis (EFA) was 10 to 15 per initial item. In this study, 430–645 subjects were required. Considering a valid response rate of approximately 30%, 1,433–2,150 questionnaires had to be distributed. The sample of 1,995 utilized in this study can be considered adequate.

## Measurement

**Demographic data.** The following demographic characteristics of the participants were examined: gender, years of psychiatric experience, advanced practice registered nurse (APRN) status (certified nurse or certified nurse specialist), experience as a psychiatric home-visiting nurse, experience in providing psychiatric outpatient care, experience in somatic care wards, and educational level. Further, the following characteristics of the hospitals or wards in which they were working were asked: hospital establishment, adoption of primary nursing, whether or not pre-discharge conferences were usually held, the participation of patients' families in the pre-discharge conferences, and the participation of multidisciplinary teams in the pre-discharge conferences. Only participants who answered that "pre-discharge conferences were usually held" responded to the question if the patients' families attended the pre-discharge conferences. Similarly, only participants who explained that "pre-discharge conferences were usually held" responded to the question of whether the conference member consisted only of hospital staff or consisted of both community and hospital staff.

**The in-hospital nursing care leading to reduction in early readmission among patients with schizophrenia scale (IRERSS).** A 43-item IRERSS was developed in this study. In the IRERSS, respondents were required to recall a patient with schizophrenia who had previously been readmitted within 90 days of discharge, but who could live in a community for more than 90 days after receiving an in-hospital intervention. The respondents answered to what extent they provided the nursing care indicated in the questionnaire during the patient's in-hospital intervention. To minimize recall bias, a diagram representing the time axis was provided with the questionnaire (see S1 Fig), and the items were expressed in concrete contents. It was easy for the respondents to recall patients who had experienced readmissions within 90 days of discharge, namely, early readmissions, because the Japanese public health insurance program defines hospitalization within 90 days of the discharge as a readmission and after more than 90 days after discharge as a new hospitalization [43]. The nursing care delivered during the above in-hospital intervention would be considered "in-hospital nursing care leading to reduction in early readmission" if patient was not re-hospitalized within 90 days following discharge after receiving the nursing care.

Responses to each item were recorded on a 5-point Likert scale that ranged from 1 (strongly disagree) to 5 (strongly agree). The items of the IRERSS were developed based on Maki et al.'s findings [44]. In their qualitative study, they examined "in-hospital nursing care leading to reduction in early readmission" among patients by conducting interviews with 17 proficient psychiatric nurses. They revealed 38 concepts of nursing practice leading to reduction in early readmission among patients with schizophrenia. Based on the 38 concepts, we created 38 items of in-hospital nursing care leading to reduction in early readmission among patients

with schizophrenia. Furthermore, five of the items (Items 39, 40, 41, 42, and 43) were added based on the feedback provided by the five experts (i.e., two researchers in the field of mental health nursing, one psychiatrist, and two experienced nursing researchers). To make the expression of the items clearer and more concise, they were revised based on the opinions of a five-person panel (i.e., two graduate students, one university faculty member, and two psychiatric nurses). Accordingly, the IRERSS assesses the degree of implementation of "in-hospital nursing care leading to reduction in early readmission among patients with schizophrenia."

**The Japanese version of the Essen climate evaluation schema (EssnCES-JPN).** EssenCES is a 17-item questionnaire that measures three aspects of the social climate in psychiatric wards (PC: Patients' cohesion and mutual support, ES: Experienced safety, and TH: Therapeutic hold). Each subscale included five items, and two items were not included in any of the subscales. Each item used a 5-point Likert scale from 0 (not at all) to 4 (very much). Higher scores indicated that respondents perceived the hospital ward climate as more positive. "Patients' cohesion and mutual support" reflected an essential quality of therapeutic communities and effectively working treatment groups [45]. "Experienced safety" referred to the level of perceived tension and threat of aggression or violence [46]. "Therapeutic hold" indicated the extent to which the unit was perceived as supportive of patients' therapeutic needs [47]. The questionnaire was originally developed to assess the social and therapeutic atmosphere of forensic psychiatric wards [48]. This questionnaire's transferability to general psychiatric settings was confirmed [46]. The Japanese version of this scale had good internal consistency and construct validity [49, 50].

Nursing performance was influenced by not only nurses' individual factors but also environmental factors [39, 51]. This study investigated the relevance between in-hospital nursing care leading to reduction in early readmission and the environmental factors, using EssenCES-JPN.

**Nursing excellence scale in clinical practice (NES).** Higher scores were indicative of self-reported nursing excellence in clinical practice. The NES consists of 35 items and seven subscales, namely: (1) collecting and using client's information continuously and efficiently, (2) performing with appropriate knowledge/skills in clinical settings, (3) developing relationships with clients/families through communication, (4) overcoming difficult conditions of clients and/or in the work environment, (5) identifying potential problems for clients and solving them creatively, (6) protecting the personality and human dignity of clients, and (7) being aware of fulfilling a number of roles as a medical and nursing team member and being able to perform them. The NES had good reliability and validity [52, 53]. The NES had similarities with the IRERSS in that nurses could easily assess their own nursing practice. To control for self-assessment bias, the questionnaire instructions explained that the content of the responses to this survey would not affect the evaluation of the participants' performance. The NES, which measures nurses' individual nursing practice, was used as one of the predictors of the IRERSS.

## Data analysis

All statistical analyses were conducted using IBM SPSS Statistics version 25. Questionnaires that contained missing responses were excluded. Descriptive analyses were performed to examine the demographic characteristics of the participants and the features of the hospitals/wards in which they worked. Participants' years of psychiatric experience were classified into three categories: less than 5 years, 5–14 years, 15 years or more. The normality of data was evaluated by means of a visual inspection of histograms and QQ-plots. All statistical tests were two-tailed and $p < .05$ was considered significant.

**Latent structure of the IRERSS.**   First, item analyses were performed on the items of the IRERSS. Ceiling and floor effects were examined for each item of the IRERSS. Inter-item and item-total correlations were computed. In the inter-item correlation analyses, items with a correlation < .30 with all items, or items with a correlation > .90 with any item were removed to avoid the risk of multicollinearity [54]. In the item-total correlation analyses, an item was excluded if the correlation between the item score and total score without the respective item was < .30.

Second, an EFA was conducted using maximum likelihood extraction and promax rotation. The Kaiser-Guttman criterion (eigenvalues > 1) was used to determine the number of factors that should be retained. Items with loadings that were lower than (i.e., < .40) onto all factors were excluded. Additionally, items that were strongly loaded (i.e., > .40) onto two or more factors were also excluded. The analysis was repeated until all the items were strongly loaded onto only a single factor. The adequacy of the EFA was evaluated by the Kaiser-Meyer-Olkin (KMO) statistic and Bartlett's test of sphericity. The internal consistency of the scale was examined by computing Cronbach's alpha values for the overall scale and each subscale of the IRERSS. The factors identified through the EFA were considered subscales within the overall scale. Each subscale score was calculated by summing the included items.

**Predictors of the scores on the IRERSS.**   Correlation and univariate analyses (i.e., unpaired t-tests, and one-way analyses of variance with post hoc analyses) were performed to examine the data obtained. Correlations between the overall scale and each subscale's score on the IRERSS, overall score on the NES, and each subscale's score on the Essen-CES-JPN were computed. Using unpaired t-tests, differences in the overall scale and each subscale of the IRERSS were compared between groups dichotomized based on each of the following variables: gender, APRN status, experience as a psychiatric home visiting nurse, experience in providing psychiatric outpatient care, experience in somatic care wards, and adoption of primary nursing. Using one-way analyses of variance and post hoc analyses (Bonferroni correction), differences in the overall scale and each subscale of the IRERSS were compared among the three groups classified using each of the following variables: years of psychiatric experience, educational level, hospital establishment, participation of families in pre-discharge conferences, and participation of multidisciplinary teams in pre-discharge conferences.

Forward-backward stepwise multiple regression analyses were conducted to control for confounding variables (inclusion value = .05 and exclusion value = .10). In the stepwise multiple regression analyses, the overall scale, and each subscale scores on the IRERSS were entered as dependent variables. The variables for which the correlation analyses and univariate analyses yielded significant results were entered as independent variables.

## Ethical considerations

Ethical approval to conduct this study was granted by the ethics committee of the Graduate School of Medicine, Nagoya University, Japan (No: 17–155). The participants were informed about the aims of the study and the benefits and risks of participation through printed forms. The participants provided written informed consent. They responded to each questionnaire anonymously, enclosed the completed questionnaires in sealed envelopes, and returned them to the researchers. Hospital directors and nurse managers did not participate in the data collection process. If participants had any questions about the study, they were able to call or email the principal investigator, whose phone number and address were listed on the documents for research cooperation. Permissions to use the EssenCES-JPN and the NES were obtained from the copyright holders.

## Result

Data were collected from 823 RNs (response rate = 41.25%). Excluding missing responses, the final sample size was 724 (valid response rate = 36.29%). The demographic characteristics of the participants and the features of the hospitals/wards in which they worked are presented in Table 1.

**Table 1. Characteristics of the participants and features of the hospitals/wards in which they were working (n = 724).**

| | | | n | % |
|---|---|---|---|---|
| **Participants** | **Gender** | | | |
| | | Male | 324 | 44.8 |
| | | Female | 400 | 55.2 |
| | **Years of psychiatric experience** | | | |
| | | < 5 | 191 | 26.4 |
| | | 5–14 | 323 | 44.6 |
| | | ≥ 15 | 210 | 29.0 |
| | **APRN status[a]** | | | |
| | | No | 698 | 96.4 |
| | | Yes | 26 | 3.6 |
| | **Experience as a psychiatric home visiting nurse** | | | |
| | | No | 555 | 76.7 |
| | | Yes | 169 | 23.3 |
| | **Experience in providing psychiatric outpatient care** | | | |
| | | No | 652 | 90.1 |
| | | Yes | 72 | 9.9 |
| | **Experience in somatic care wards** | | | |
| | | No | 264 | 36.5 |
| | | Yes | 460 | 63.5 |
| | **Educational level** | | | |
| | | Diploma | 598 | 82.6 |
| | | Bachelor | 112 | 15.5 |
| | | Master | 14 | 1.9 |
| **Hospitals/ Wards** | **Hospital establishment** | | | |
| | | Public | 366 | 50.6 |
| | | Non-profit corporation | 27 | 3.7 |
| | | Private | 331 | 45.7 |
| | **Adoption of primary nursing** | | | |
| | | No | 325 | 44.9 |
| | | Yes | 399 | 55.1 |
| | **Holding pre-discharge conferences** | | | |
| | | No | 166 | 2.9 |
| | | Yes | 558 | 77.1 |
| | **Participation of families in predischarge conferences** | | | |
| | | No | 122 | 16.9 |
| | | Yes | 436 | 60.2 |
| | **Participation of multidisciplinary teams in predischarge conferences** | | | |
| | | Consist of hospital staff only | 312 | 43.1 |
| | | Consist of both community care provider and hospital staff member | 246 | 34.0 |

[a] Certified nurse or certified nurse specialist.

## Latent structure of the IRERSS

No ceiling or floor effects were observed for any of the items of the IRERSS. Based on the results of inter-item correlation analyses, Item 3 was excluded because its coefficients with all items were lower than .30. None of the item-total correlations of the items were lower than .30. The descriptive statistics of the IRERSS are shown in S1 Table.

In the EFA, six items (Items 1, 2, 19, 20, 27, and 36) were excluded because their loadings onto all factors were lower than .40 or cross-loaded (loadings > .40) onto two or more factors. The final model of the IRERSS was a 5-factor structure that consisted of 36 items. Further, the KMO statistic was .96, and Bartlett's test of sphericity yielded significant results ($\chi^2$ = 14 772, df = 630, p < .01). The collected data were considered superior [42]. Table 2 presents the results of the EFA of the 36-item IRERSS. Factor 1 was labeled "Promoting cognitive functioning and self-care" and consisted of nine items. Factor 2 was labeled "Identifying reasons for readmission" and consisted of eight items. Factor 3 was labeled "Establishing cooperative systems within the community" and consisted of seven items. Factor 4 was labeled "Sharing goals about community life" and consisted of seven items. Factor 5 was labeled "Creating restful spaces" and consisted of five items. Their Cronbach's alphas ranged from .81 to .91 (see Table 2). The mean overall score of the IRERSS was 131.92 (SD = 18.09). The Cronbach's alpha of the 36-item IRERSS was .96.

**Predictors of the scores on the IRERSS.**   The mean scores of the overall NES, "Patients' cohesion and mutual support", "Experienced safety", and "Therapeutic hold" were 121.65 (SD = 18.42), 10.06 (SD = 2.57), 6.48 (SD = 3.82), and 12.58 (SD = 2.93), respectively. Table 3 shows the correlation matrix for the overall scale and each subscale's score on the IRERSS, each subscale's score on the EssenCES-JPN, and the overall score on the NES. The correlation obtained between the overall score on the IRERSS and the overall score on the NES was r = .59, p < .01; between the overall score on the IRERSS and the score on the "Patients' cohesion and mutual support" was r = .18, p < .01; between the overall score on the IRERSS and the score on the "Experienced safety" was r = -.07, p = .06; and between the overall score on the IRERSS and the score on the "Therapeutic hold" was r = .33, p < .01.

The results of the univariate analyses, namely, the unpaired t-tests, and one-way analyses of variance and the post hoc analyses (Bonferroni correction), on the overall scale score and each subscale's score on the IRERSS are summarized in Table 4. Significant differences were found in the univariate analyses with the overall score on the IRERSS as a dependent variable and the following nine variables as independent variables: years of psychiatric experience, APRN status, experience as a psychiatric home visiting nurse, experience in providing psychiatric outpatient care, educational level, hospital establishment, the adoption of primary nursing, the participation of families in pre-discharge conferences, and the participation of multidisciplinary teams in pre-discharge conferences.

In a stepwise multiple regression analysis with the overall score on the IRERSS as the dependent variable, the following 12 variables were entered as independent variables: years of psychiatric experience, APRN status, experience as a psychiatric home visiting nurse, experience in providing psychiatric outpatient care, educational level, hospital establishment, the adoption of primary nursing, the participation of families in pre-discharge conferences, the participation of multidisciplinary teams in pre-discharge conferences, the score on "Patients' cohesion and mutual support", the score on "Therapeutic hold", and the overall score on the NES. The stepwise multiple regression analyses revealed that the overall scale score on the IRERSS was predicted by the following five variables: the overall score on the NES ($\beta$ = .53, p < .01), the score on "Therapeutic hold" ($\beta$ = .12, p < .01), APRN status ($\beta$ = .09, p < .01), the participation of multidisciplinary teams in pre-discharge conferences ($\beta$ = .08, p = .01), and

**Table 2. Exploratory factor analyses and internal consistency of the IRERSS (n = 724).**

| Total: IRERSS (α = .958) | | Factor loadings | | | | | h² a |
|---|---|---|---|---|---|---|---|
| | | F1 | F2 | F3 | F4 | F5 | |
| Factor 1: Promoting cognitive functioning and self-care (α = .908) | | | | | | | |
| #30 | I talked with the patient about how to deal with delusions so that he/she could take responsibility for behaviors. | **.960** | -.072 | -.075 | .035 | -.137 | .648 |
| #29 | I helped the patient reconsider his/her thoughts so that he/she could realize that his/her delusions were thoughts that were inconsistent with reality. | **.926** | -.071 | -.047 | .048 | -.183 | .591 |
| #33 | I observed how the patient deals with delusions. | **.741** | -.012 | -.013 | -.024 | .057 | .564 |
| #31 | I evaluated my nursing care from changes in the patient's behaviors. | **.676** | .071 | .055 | -.025 | -.037 | .514 |
| #32 | I believed in the patient and encourage him/her to change behaviors. | **.592** | .126 | .000 | -.056 | .088 | .503 |
| #34 | I helped the patient understand the need for medication by providing factual information. | **.581** | -.002 | .067 | .012 | .122 | .524 |
| #35 | I tried to notice changes in the patient's attitudes toward medication | **.554** | .119 | .087 | -.051 | .101 | .544 |
| #28 | I helped the patient improve his/her lifestyle. | **.493** | -.001 | .073 | .190 | .064 | .540 |
| #26 | I helped the patient accept his/her disability. | **.427** | .120 | .052 | .019 | .137 | .453 |
| Factor 2: Identifying reasons for readmission (α = .883) | | | | | | | |
| #9 | I tried to understand the patient's capabilities and the challenges that he/she had been facing. | -.025 | **.828** | .036 | .028 | -.104 | .616 |
| #8 | I tried to gain a more detailed understanding about why the patient was readmitted. | -.036 | **.738** | .029 | -.022 | -.001 | .510 |
| #7 | I tried to understand the challenges that the patient had been facing based on my observations of his/her daily life. | .139 | **.723** | -.024 | -.046 | -.058 | .532 |
| #6 | As a nurse, I paid close attention to the things about which the patient was worried. | .085 | **.718** | -.050 | -.106 | .075 | .525 |
| #10 | I envisioned nursing goals that aimed to enhance the post-discharge well-being of the patient. | -.034 | **.660** | .019 | .268 | -.113 | .623 |
| #11 | I created nursing plans that incorporated the opinions of other nursing staff members. | -.051 | **.578** | .039 | .171 | .010 | .508 |
| #5 | I shared important information about the patient with other nursing staff members. | -.035 | **.536** | -.055 | -.064 | .223 | .371 |
| #4 | I often cooperated with other nursing staff members. | -.048 | **.430** | .052 | .163 | .047 | .359 |
| Factor 3: Establishing cooperative systems within the community (α = .882) | | | | | | | |
| #42 | I discussed with the patient the services that he/she wanted to use in his/her community. | .077 | -.074 | **.845** | -.044 | -.073 | .629 |
| #40 | I participated in care conferences that involved community care providers. | -.140 | .032 | **.844** | .064 | -.096 | .606 |
| #43 | I informed the patient and his/her family about the support system that would be available to the patient after discharge. | .068 | -.019 | **.760** | -.097 | -.101 | .654 |
| #41 | I assessed the patient's self-care abilities by comparing his/her behaviors at the time of admission and discharge. | .127 | .040 | **.734** | -.122 | -.038 | .548 |
| #38 | I put the patient in touch with community nurses to ensure the continuity of care. | -.155 | .042 | **.685** | .032 | .129 | .503 |

(*Continued*)

**Table 2.** (Continued)

| Total: IRERSS ($\alpha$ = .958) | | Factor loadings | | | | | $h^{2\,a}$ |
|---|---|---|---|---|---|---|---|
| | | F1 | F2 | F3 | F4 | F5 | |
| #37 | I was in touch with the patient and his/her caregivers, for a while, even after discharge. | .021 | .002 | **.497** | .062 | .209 | .486 |
| #39 | I discussed with the patient the good things that could happen to him/her after discharge with him/her. | .209 | -.006 | **.430** | -.038 | .097 | .382 |
| **Factor 4: Sharing goals about community life** ($\alpha$ = .891) | | | | | | | |
| #15 | I discussed with the patient and his/her family how he/she can adapt to community life. | -.068 | .034 | .051 | **.840** | -.081 | .648 |
| #14 | I discussed with the patient and his/her family what he/she wanted to do in his/her community. | .032 | .000 | .021 | **.789** | -.096 | .581 |
| #17 | I helped the patient prepare for community life. | .052 | -.080 | .059 | **.699** | .087 | .606 |
| #16 | I helped the patient practice what he/she was not good at. | .065 | -.004 | -.018 | **.668** | .105 | .573 |
| #18 | I reassured the patient and his/her family about his/her ability to adapt to community life after discharge. | .065 | -.004 | -.018 | **.645** | .110 | .571 |
| #13 | I relieved the anxiety that the patient and his/her family experienced about community life. | .096 | .077 | .031 | **.550** | -.090 | .407 |
| #12 | I valued the happiness of the patient and his/her family. | .044 | .202 | -.080 | **.549** | -.005 | .473 |
| **Factor 5: Creating restful spaces** ($\alpha$ = .810) | | | | | | | |
| #23 | I created spaces within which the patient did not feel stressed. | -.099 | -.022 | -.040 | -.114 | **.890** | .524 |
| #22 | I allowed the patient to take rest and calm himself/herself down. | .119 | .049 | -.119 | .063 | **.589** | .484 |
| #24 | I shared nursing goals with other nursing staff members. | -.071 | .024 | .073 | .131 | **.575** | .464 |
| #21 | I assured the patient that hospitals are safe spaces. | .156 | .108 | -.005 | .086 | **.438** | .498 |
| #25 | I helped the patient feel more hopeful about community life. | .180 | .008 | .051 | .087 | **.409** | .429 |
| | Eigenvalues | 14.8 | 2.2 | 2.0 | 1.2 | 1.2 | |
| | Inter-factor correlation | F1 | 1 | | | | | |
| | | F2 | .644 | 1 | | | | |
| | | F3 | .632 | .554 | 1 | | | |
| | | F4 | .655 | .749 | .672 | 1 | | |
| | | F5 | .707 | .697 | .522 | .677 | 1 | |

Abbreviations: IRERSS = In-hospital nursing care leading to reduction in early readmission among patients with schizophrenia scale.

[a] $h^2$: Communalities.

the participation of families in pre-discharge conferences ($\beta$ = .07, p = .03). The results of the multiple stepwise regression analyses are presented in Table 5.

## Discussion

The main findings of this study can be summarized as follows: the IRERSS was found to consist of five factors and 36 items, and the overall scores of the IRERSS were significantly associated with APRN status, the overall score on the NES, the score on "Therapeutic hold", the

**Table 3. Correlation between the overall scale and each subscale score on the IRERSS, each subscale score of the EssenCES-JPN, and the overall score on the NES (n = 724).**

| Scales | NES | PC | ES | TH |
|---|---|---|---|---|
| **IRERSS total score** | .590** | .180** | -.071 | .331** |
| **Factor 1:** Promoting cognitive functioning and self-care | .534** | .152** | -.095* | .260** |
| **Factor 2:** Identifying reasons for readmission | .547** | .119** | -.033 | .315** |
| **Factor 3:** Establishing cooperative systems within the community | 414** | .182** | -.076* | .277** |
| **Factor 4:** Sharing goals about community life | .472** | .160** | -.037 | .257** |
| **Factor 5:** Creating restful spaces | .512** | .133** | -.039 | .284** |

Abbreviations: IRERSS = In-hospital nursing care leading to reduction in early readmission among patients with schizophrenia scale, NES = Nursing excellence scale in clinical practice, PC = Patients' cohesion and mutual support, ES = Experienced safety, TH = Therapeutic hold.

*p < .01

**p < .05.

participation of families in pre-discharge conferences, and the participation of multidisciplinary teams in pre-discharge conferences. To the best of our knowledge, this is the first study to investigate the latent structure and predictors of in-hospital nursing care leading to reduction in early readmission among patients with schizophrenia.

## Latent structure of the IRERSS

The item analyses made it possible to avoid collinearity of items in the IRERSS and ensure sufficient internal consistency. Item 3, which was excluded in the inter-item correlation analysis on the 43-item IRERSS, was not included in "In-hospital nursing care leading to reduction in early readmission", because the correlation coefficients between Item 3 and all the other items were less than .30.

The five-factor structure of the IRERSS was largely consistent with the findings of the previous study [44]. The previous study revealed qualitatively that hospital nurses' practices leading to reduction in early readmission among patients with schizophrenia could be classified into five categories. Each factor of the IRERSS largely overlapped with each of Maki et al.'s five categories as follows [44]: Factor 1 of the IRERSS (i.e., Promoting cognitive functioning and self-care), Category 1 in the previous study (i.e., Supporting the patient so that he/she can accept his/her illness); Factor 2 (i.e., Identifying reasons for readmission), Category 2 (i.e., As a sympathizer, understanding the patient's tasks and predicting the post-discharge life of each patient); Factor 3 (i.e., Establishing cooperative systems within the community), Category 3 (i.e., Bridging the gap between the community and the hospital, including nursing in the hospital, as part of the process); Factor 4 (i.e., Sharing goals about community life), Category 4 (i.e., Mediating between the patient and the family, with consideration for the happiness of both the patient and his/her family after discharge); Factor 5 (i.e., Creating restful spaces), Category 5 (i.e., Establishing a relationship of trust with the patient, ensuring the patient's sense of security, and considering his/her impaired ability to establish rapport). This study showed that Japanese in-hospital nursing care leading to reduction in early readmission among patients with schizophrenia comprised these five factors.

The five factors of the IRERSS include several elements that may influence the risk of readmission. Most studies reported that the existence or the number of previous admissions was

**Table 4. Univariate analyses of the IRERSS (n = 724).**

| | | Total: IRERSS | | Factor 1: Promoting cognitive functioning and self-care | | Factor 2: Identifying reasons for readmission | | Factor 3: Establishing cooperative systems within the community | | Factor 4: Sharing goals about community life | | Factor 5: Creating restful spaces | |
|---|---|---|---|---|---|---|---|---|---|---|---|---|---|
| | n | Mean | (SD) | Mean | (SD) | Mean | (SD) | Mean | (SD) | Mean | (SD) | Mean | (SD) |
| **Gender[a]** | | | | | | | | | | | | | |
| Male | 324 | 131.6 | (17.7) | 32.5 | (5.1) | 30.0 | (4.4) | 24.6 | (4.6) | 25.7 | (4.1) | 18.9 | (2.6) |
| Female | 400 | 132.2 | (18.4) | 32.2 | (5.4) | 30.6 | (4.6) | 24.7 | (4.1) | 25.7 | (4.3) | 18.9 | (2.8) |
| | | t = -.38 | | t = .67 | | t = -1.82 | | t = -.47 | | t = -.17 | | t = .29 | |
| | | p = .71 | | p = .50 | | p = .07 | | p = .64 | | p = .87 | | p = .78 | |
| **Years of psychiatric experience[b]** | | | | | | | | | | | | | |
| < 5 | 191 | 128.1 | (17.3)[c] | 31.4 | (5.0)[c] | 29.6 | (4.3)[c] | 23.8 | (4.9)[c] | 24.9 | (4.3)[c] | 18.5 | (2.6)[c] |
| 5–14 | 323 | 131.1 | (17.9)[d] | 32.1 | (5.4)[d] | 30.2 | (4.5)[f] | 24.4 | (4.7)[d] | 25.6 | (4.1)[f] | 18.8 | (2.7) |
| ≥ 15 | 210 | 136.6 | (18.2)[c, d] | 33.5 | (5.3)[c, d] | 31.3 | (4.5)[c,f] | 25.9 | (4.9)[c, d] | 26.6 | (4.0)[c, f] | 19.4 | (2.8)[c] |
| | | F = 12.06 | | F = 8.32 | | F = 7.40 | | F = 10.45 | | F = 8.89 | | F = 6.34 | |
| | | p < .01 | | p < .01 | | p < .01 | | p < .01 | | p < .01 | | p < .01 | |
| **APRN status[a]** | | | | | | | | | | | | | |
| No | 698 | 131.4 | (18.0) | 32.2 | (5.3) | 30.2 | (4.5) | 24.6 | (4.9) | 25.6 | (4.2) | 18.8 | (2.7) |
| Yes | 26 | 145.9 | (16.0) | 35.4 | (4.5) | 33.5 | (3.2) | 27.3 | (4.8) | 28.9 | (4.1) | 20.9 | (2.6) |
| | | t = -4.06 | | t = -3.07 | | t = -3.68 | | t = -2.78 | | t = -3.94 | | t = -3.85 | |
| | | p < .01 | | p < .01 | | p < .01 | | p < .01 | | p < .01 | | p < .01 | |
| **Experience as a psychiatric home visiting nurse[a]** | | | | | | | | | | | | | |
| No | 555 | 130.5 | (17.9) | 32.0 | (5.2) | 30.2 | (4.4) | 24.2 | (4.9) | 25.5 | (4.3) | 18.8 | (2.7) |
| Yes | 169 | 136.5 | (18.1) | 33.5 | (5.4) | 30.9 | (4.6) | 26.2 | (4.6) | 26.5 | (3.9) | 19.3 | (2.8) |
| | | t = -3.76 | | t = -3.33 | | t = -1.88 | | t = -4.73 | | t = -2.93 | | t = -2.39 | |
| | | p < .01 | | p < .01 | | p = .06 | | p < .01 | | p < .01 | | p = .02 | |
| **Experience in providing psychiatric outpatient care[a]** | | | | | | | | | | | | | |
| No | 652 | 131.4 | (18.0) | 32.3 | (5.3) | 30.2 | (4.5) | 24.5 | (4.9) | 25.6 | (4.2) | 18.9 | (2.7) |
| Yes | 72 | 136.2 | (18.0) | 32.9 | (5.6) | 31.9 | (4.2) | 25.7 | (4.7) | 26.5 | (4.3) | 19.2 | (2.8) |
| | | t = -2.12 | | t = -1.02 | | t = -3.20 | | t = -1.95 | | t = -1.62 | | t = -.86 | |
| | | p = .03 | | p = .31 | | p < .01 | | p = .05 | | p = .11 | | p = .39 | |
| **Experience in somatic care wards[a]** | | | | | | | | | | | | | |
| No | 264 | 132.0 | (17.7) | 32.5 | (5.0) | 30.5 | (4.1) | 24.6 | (4.8) | 25.7 | (4.3) | 18.8 | (2.7) |
| Yes | 460 | 131.9 | (18.3) | 32.2 | (5.5) | 30.3 | (4.7) | 24.7 | (4.9) | 25.7 | (4.1) | 18.9 | (2.7) |
| | | t = .13 | | t = .85 | | t = -.55 | | t = -.38 | | t = -.25 | | t = -.63 | |
| | | p = .90 | | p = .40 | | p = .58 | | p = .70 | | p = .80 | | p = .53 | |
| **Education level[b]** | | | | | | | | | | | | | |
| Diploma | 598 | 131.4 | (18.1)[f] | 32.2 | (5.4)[f] | 30.2 | (4.5) | 24.6 | (4.9) | 25.6 | (4.1) | 18.8 | (2.7)[f] |
| Bachelor | 112 | 133.3 | (17.5) | 32.6 | (4.7) | 30.7 | (4.5) | 24.8 | (4.8) | 26.1 | (4.6) | 19.2 | (2.8) |
| Master | 14 | 143.6 | (19.0)[f] | 35.9 | (5.1)[f] | 32.6 | (4.2) | 26.9 | (4.9) | 27.7 | (4.8) | 20.6 | (2.8)[f] |
| | | F = 3.57 | | F = 3.47 | | F = 2.50 | | F = 1.50 | | F = 2.25 | | F = 3.86 | |
| | | p = .03 | | p = .03 | | p = .08 | | p = .22 | | p = .11 | | p = .02 | |
| **Hospital establishment[b]** | | | | | | | | | | | | | |
| Public | 366 | 134.3 | (17.5)[c] | 32.9 | (5.3)[c] | 30.9 | (4.2)[c] | 24.9 | (4.9) | 26.2 | (4.1)[c] | 19.3 | (2.7)[c] |
| Non-profit corporation | 27 | 129.8 | (19.1) | 31.5 | (6.0) | 29.6 | (3.9) | 24.6 | (5.0) | 25.6 | (4.6) | 18.5 | (2.7) |
| Private | 331 | 129.5 | (18.3)[c] | 31.7 | (5.1)[c] | 29.8 | (4.7)[c] | 24.4 | (4.9) | 25.1 | (4.2)[c] | 18.5 | (2.7)[c] |
| | | F = 6.57 | | F = 4.96 | | F = 6.54 | | F = 1.26 | | F = 6.17 | | F = 7.38 | |
| | | p < .01 | | p < .01 | | p < .01 | | p = .28 | | p < .01 | | p < .01 | |
| **Adoption of primary nursing[a]** | | | | | | | | | | | | | |

(*Continued*)

**Table 4.** (*Continued*)

| | n | Total: IRERSS | | Factor 1: Promoting cognitive functioning and self-care | | Factor 2: Identifying reasons for readmission | | Factor 3: Establishing cooperative systems within the community | | Factor 4: Sharing goals about community life | | Factor 5: Creating restful spaces | |
|---|---|---|---|---|---|---|---|---|---|---|---|---|---|
| | | Mean | (SD) | Mean | (SD) | Mean | (SD) | Mean | (SD) | Mean | (SD) | Mean | (SD) |
| No | 325 | 128.8 | (17.8) | 31.6 | (5.2) | 29.7 | (4.7) | 23.9 | (4.9) | 25.1 | (4.1) | 18.5 | (2.7) |
| Yes | 399 | 134.4 | (17.9) | 32.9 | (5.3) | 30.9 | (4.3) | 25.3 | (4.8) | 26.2 | (4.3) | 19.2 | (2.7) |
| | | t = -4.21 | | t = -3.14 | | t = -3.81 | | t = -3.69 | | t = -3.65 | | t = -3.23 | |
| | | $p < .01$ | | $p < .01$ | | $p < .01$ | | $p < .01$ | | $p < .01$ | | $p < .01$ | |
| **Participation of families in pre-discharge conferences**[b] | | | | | | | | | | | | | |
| Not hold pre-discharge conferences | 166 | 128.1 | (18.3)[c] | 32.1 | (5.3) | 29.6 | (4.8)[f] | 23.0 | (5.1) | 24.8 | (4.5)[c] | 18.6 | (2.8) |
| No | 122 | 129.2 | (19.5)[f] | 32.1 | (5.9) | 29.9 | (4.7) | 23.7 | (5.0)[d] | 24.7 | (4.4)[d] | 18.7 | (2.8) |
| Yes | 436 | 134.1 | (17.3)[c, f] | 32.5 | (5.1) | 30.8 | (4.2)[f] | 25.6 | (4.5)[c, d] | 26.3 | (3.9)[c, d] | 19.0 | (2.7) |
| | | F = 8.55 | | F = .461 | | F = 4.82 | | F = 20.07 | | F = 12.37 | | F = 1.54 | |
| | | $p < .01$ | | p = .63 | | $p < .01$ | | $p < .01$ | | $p < .01$ | | p = .22 | |
| **Participation of multidisciplinary teams in pre-discharge conferences**[b] | | | | | | | | | | | | | |
| Not hold pre-discharge conferences | 166 | 128.1 | (18.3)[c] | 32.1 | (5.3) | 29.6 | (4.8)[c, d] | 23.0 | (5.1)[c, d] | 24.8 | (4.5)[c] | 18.6 | (2.8) |
| Only hospital staff | 312 | 131.3 | (17.7)[f] | 32.2 | (5.4) | 30.2 | (4.2)[c e] | 24.4 | (4.7)[c e] | 25.6 | (4.0) | 19.0 | (2.7) |
| Community and hospital staff members | 246 | 135.2 | (18.0)[c, f] | 32.7 | (5.2) | 31.1 | (4.5)[d, e] | 26.1 | (4.5)[d, e] | 26.4 | (4.1)[c] | 19.0 | (2.7) |
| | | F = 8.14 | | F = .85 | | F = 6.14 | | F = 21.83 | | F = 7.31 | | F = .98 | |
| | | $p < .01$ | | p = .43 | | $p < .01$ | | $p < .01$ | | $p < .01$ | | p = .38 | |

Abbreviations: IRERSS = In-hospital nursing care leading to reduction in early readmission among patients with schizophrenia scale, SD = standard deviation,

APRN = advanced practice registered nurse (certified nurse or certified nurse specialist).

[a]: unpaired t-tests.

[b]: one-way analyses of variance, and post-hoc analyses (Bonferroni correction).

[c, d, e]$p < .01$ (a significant difference was observed between groups).

[f, g]$p < .05$ (a significant difference was observed between groups).

associated with the risk of readmission [12]. Factor 2 of the IRERSS (i.e., Identifying reasons for readmission) can be useful for finding the reasons for patients' past hospitalizations and can be linked to the care leading to reduction in early readmission. For example, Item 9 of the IRERSS (i.e., I tried to understand the patient's capabilities and the challenges that he/she had been facing), and Item 8 of the IRERSS (i.e., I tried to gain a more detailed understanding about why the patient was readmitted), could help to identify reasons for past hospitalizations. Self-management education interventions were associated with a significant reduction of re-hospitalization and more likely to improve patients' adherence to medication [55]. Factor 1 of the IRERSS (i.e., Promoting cognitive functioning and self-care) had much in common with self-management education interventions. In self-management education, patients learn problem solving skills which allow them to take appropriate actions to improve their health [56]. Item 30 of the IRERSS (i.e., I talked with the patient about how to deal with delusions so that he/she could take responsibility for behaviors), and Item 29 of the IRERSS (i.e., I helped the patient reconsider his/her thoughts so that he/she could realize that his/her delusions were thoughts that were inconsistent with reality) may be effective for patients to acquire behaviors to improve their health.

**Table 5. Stepwise multiple regressions of the IRERSS (n = 724).**

| | Total: IRERSS | | Factor 1: Promoting cognitive functioning and self-care | | Factor 2: Identifying reasons for readmission | | Factor 3: Establishing cooperative systems within the community | | Factor 4: Sharing goals about community life | | Factor 5: Creating restful spaces | |
|---|---|---|---|---|---|---|---|---|---|---|---|---|
| | B [95% CI] | β | B [95% CI] | β | B [95% CI] | β | B [95% CI] | β | B [95% CI] | β | B [95% CI] | β |
| **APRN status[a]** | | | | | | | | | | | | |
| No | Reference | | | | Reference | | | | Reference | | Reference | |
| Yes | 8.36 [2.76, 13.96] | .09** | | | 1.85 [.40, 3.30] | .08* | | | 2.10 [.68, 3.53] | .09** | 1.27 [.36, 2.18] | .09** |
| | VIF = 1.01 | | | | VIF = 1.01 | | | | VIF = 1.01 | | VIF = 1.01 | |
| **Experience as a psychiatric home visiting nurse** | | | | | | | | | | | | |
| No | | | | | | | Reference | | | | | |
| Yes | | | | | | | 1.26 [.52, 2.00] | .11** | | | | |
| | | | | | | | VIF = 1.03 | | | | | |
| **Education level** | | | | | | | | | | | | |
| Diploma | | | Reference | | | | | | | | | |
| Bachelor | | | | | | | | | | | | |
| Master | | | .15 [.03, .27] | .08* | | | | | | | | |
| | | | VIF = 1.03 | | | | | | | | | |
| **Participation of families in pre-discharge conferences** | | | | | | | | | | | | |
| Not hold the conferences | Reference | | | | | | Reference | | Reference | | | |
| No | | | | | | | | | | | | |
| Yes | 2.61 [.25, 4.97] | .07* | | | | | 1.28 [.58, 1.98] | .13** | 1.34 [.80, 1.88] | .16** | | |
| | VIF = 1.24 | | | | | | VIF = 1.25 | | VIF = 1.00 | | | |
| **Participation of multidisciplinary teams in pre-discharge conferences** | | | | | | | | | | | | |
| Not hold the conferences | Reference | | | | Reference | | Reference | | | | | |
| Hospital staff member only | | | | | | | | | | | | |
| Community and hospital staff member | 3.16 [.68, 5.46] | .08* | | | .96 [.39, 1.53] | .10** | 1.41 [.69, 2.14] | .14** | | | | |
| | VIF = 1.24 | | | | VIF = 1.01 | | VIF = 1.24 | | | | | |
| **NES** | .52 [.46, .58] | .53** | .14 [.13, .16] | .50** | .12 [.10, .14] | .49** | .09 [.07, .11] | .34** | .10 [.09, .12] | .44** | .07 [.06, .08] | .46** |
| | VIF = 1.16 | | VIF = 1.15 | | VIF = 1.16 | | VIF = 1.17 | | VIF = 1.04 | | VIF = 1.16 | |
| **PC** | | | | | | | .16 [.03, .28] | .08* | .14 [.03, .24] | .08* | | |
| | | | | | | | VIF = 1.11 | | VIF = 1.03 | | | |
| **ES** | | | -.122 [-.21, -04] | -.09** | | | | | | | | |
| | | | VIF = 1.00 | | | | | | | | | |
| **TH** | .77 [.39, 1.15] | .12** | .15 [.03, .27] | .08* | .20 [.10, .30] | .13** | .17 [.05, .28] | .10** | | | .11 [.05, .17] | .12** |
| | VIF = 1.16 | | VIF = 1.16 | | VIF = 1.16 | | VIF = 1.26 | | | | VIF = 1.14 | |
| **R²** | .389 | | .303 | | .332 | | .262 | | .263 | | .282 | |
| **Adjusted R²** | .385 | | .299 | | .328 | | .256 | | .259 | | .279 | |

Abbreviations: IRERSS = In-hospital nursing care leading to reduction in early readmission among patients with schizophrenia scale, CI = confidence interval,
APRN = advanced practice registered nurse (certified nurse or certified nurse specialist), NES = Nursing excellence scale in clinical practice, PC = Patients' cohesion and mutual support, ES = Experienced safety, TH = Therapeutic hold.

*p < .01

**p < .05.

### Predictors of the scores on the IRERSS

**The score on the NES.**   The score on the NES in this study may reflect the excellence of psychiatric hospital nurses all over Japan. The mean score on the overall NES in this study was similar to that of the previous study [52]. In the previous study, a survey using the NES was performed on 799 nurses in Japan, and the average score on the overall NES was 122.0 (SD = 18.1). This study also conducted a survey of hospital nurses across Japan. The excellence of the participants in this study might reflect that of nurses all over Japan.

The score on the overall NES may represent nurses' personal factors in their nursing performance indicated by the score on the overall IRERSS. The NES consisted of seven subscales. Each of these subscales was a personal component of nursing excellence. Nursing performance was influenced by not only individual factors but also environmental factors [39, 51]. In this study's stepwise regression analysis, the score on the overall NES was the most significant variable among predictor variables. The individual factors of nurses, which was represented by the score on the overall NES, might predict most of the scores on the overall IRERSS.

**Therapeutic ward climate.**   In Japan, nurses' perception of the social climate may be less therapeutic than the social climate perceived by nurses in European countries. The mean score on "Therapeutic hold" in this study was lower than in closed general psychiatric wards in Germany (i.e., mean = 13.8, SD was not reported) [46], lower than in open general psychiatric wards in Germany (i.e., mean = 15.6, SD was not reported) [46], and lower than in English high-security hospital settings (i.e., mean = 14.17, SD = 3.29) [57]. The average score on "Therapeutic hold" in this study was similar to that of a previous study in Japan (i.e., mean = 13.44, SD = 3.38) [50]. Japanese nurses may perceive the social climate in psychiatric wards to be less sensitive of patients' needs than foreign nurses.

The therapeutic ward climate was confirmed to be among the predictors of in-hospital nursing care leading to reduction in early readmission. Nursing performance is influenced by nurses' personal, environmental, and patients' factors [58]. The R-squared in the stepwise multiple regression analyses conducted in this study were not high. The R-squared represents the amount of variance in the dependent variable that is explained by the independent variables. It ranges from 1.0 (perfect prediction) to 0.0 (no prediction) [59]. In this study, data on variables related to patients' characteristics were not collected and they were not entered in the stepwise regression analyses. This may explain why the R-squared was not high. If the variables on patients' characteristics were entered into the stepwise regression analyses, the R-squared may have been higher. This study could reveal only that environmental factors as well as nurses' individual factors influenced in-hospital nursing care leading to reduction in early readmission among patients with schizophrenia.

**Systems that facilitate communication with community care providers.**   The present findings underscore the need to develop systems that facilitate communication between patients and community care providers such as family and multidisciplinary teams. The results of the multiple regression analyses revealed that the score on the overall IRERSS was significantly associated with the participation of families and multidisciplinary teams (both community and hospital staff members) in pre-discharge conferences. In previous studies, interventions to engage patients with community care providers were found to reduce readmissions [10, 60–62]. This study quantitatively confirmed that communication between patients and community care providers promotes in-hospital nursing care leading to reduction in early readmission among patients with schizophrenia.

### Limitations

This study has several limitations. First, the response rate in this study was low. The generalizability of our findings may be limited. The results of this study may have been influenced by

the answers of nurses who had high standards of nursing excellence and were supportive of patients' needs. The low response rate may be attributed to the high number of items in the questionnaire and their complicated wording. This suggests the need to refine the items of the IRERSS to create more straightforward sentences. Second, nurses' responses may have been influenced by recall biases. The participants may have answered desirably without total recall. The scales used in this study could have been better evaluated. The design adopted in this study could not completely eliminate the recall bias. Respondents should be required to recall more recent nursing practices, such as those within the last year. Alternatively, the questionnaire could utilize a vignette case of a fictitious patient with schizophrenia who was readmitted early. Third, this study did not investigate the data of patients, therapies that patients were offered, and the duration of hospitalization. The nursing care provided to patients depended on their mental state and social background. It is necessary to reveal the degree that the patients' characteristics influenced nursing care leading to reduction in early readmission. Fourth, this study could not clarify the difference between interventions delivered to patients who had been readmitted early and those specifically designed to prevent early readmission. In other words, the design adopted in this study could not reveal which factors or items of the IRERSS were effective in preventing early readmission of patients with schizophrenia. The IRERSS would include the interventions that are effective in preventing early readmission and those provided usually during hospitalization. Early readmission might also be affected by patients' medical state, therapies they received, and community care after discharge. Therefore, future studies using the IRERSS need to compare nursing practices implemented for patients with schizophrenia who were readmitted early and for those who were not, adjusting for the effect of patients' medical state, therapies they received, and community care after discharge.

## Conclusion

This study elucidated that Japanese nurses working in hospitals provide nursing care consisting of five factors leading to reduction in early readmission among patients with schizophrenia. In-hospital nursing care leading to reduction in early readmission among patients with schizophrenia was associated with not only nurses' individual factors but also environmental factors such as therapeutic ward climate, the participation of families in pre-discharge conferences, and the participation of multidisciplinary teams in pre-discharge conferences. To facilitate in-hospital nursing care leading to reduction in early readmission, nurses should foster a social climate in which patients feel cared for and develop systems that facilitate communication with community care providers.

## Supporting information

**S1 Dataset.**
(XLSX)

**S1 Fig. In-hospital nursing care leading to reduction in early readmission.**
(DOCX)

**S1 Table. Descriptive statistics of the IRERSS.**
(DOCX)

**S1 Appendix. 43-item IRERSS (Engilsh).**
(DOCX)

**S2 Appendix. 43-item IRERSS (Japanese).**
(DOCX)

**S3 Appendix. Questionnaire used to collect demographic information (English).**
(DOCX)

**S4 Appendix. Questionnaire used to collect demographic information (Japanese).**
(DOCX)

# Acknowledgments

We would like to thank the nurses who participated in this study, as well as the graduate students, the university faculty members, the psychiatric nurses, and the psychiatrist for participating in discussions on this research.

# Author Contributions

**Conceptualization:** Shigeyoshi Maki, Kuniyoshi Nagai, Shoko Ando, Koji Tamakoshi.

**Data curation:** Shigeyoshi Maki.

**Formal analysis:** Shigeyoshi Maki, Shoko Ando, Koji Tamakoshi.

**Funding acquisition:** Shigeyoshi Maki.

**Investigation:** Shigeyoshi Maki.

**Methodology:** Shigeyoshi Maki, Kuniyoshi Nagai, Shoko Ando, Koji Tamakoshi.

**Project administration:** Shigeyoshi Maki, Shoko Ando.

**Resources:** Shigeyoshi Maki.

**Software:** Shigeyoshi Maki.

**Supervision:** Kuniyoshi Nagai, Shoko Ando, Koji Tamakoshi.

**Validation:** Shoko Ando, Koji Tamakoshi.

**Writing – original draft:** Shigeyoshi Maki.

**Writing – review & editing:** Kuniyoshi Nagai, Shoko Ando, Koji Tamakoshi.

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
