## [Decision Letter · Decision Letter 0]

4 Feb 2021

PONE-D-20-29533

Structure and predictors of in-hospital nursing care to prevent early readmission in patients with schizophrenia in Japan: A cross-sectional study

PLOS ONE

Dear Dr. Maki,

Thank you for submitting your manuscript to PLOS ONE. After careful consideration, we feel that it has merit but does not fully meet PLOS ONE’s publication criteria as it currently stands. Therefore, we invite you to submit a revised version of the manuscript that addresses the points raised during the review process.

First, I would like to apologize for the wait between your submission and a decision. This was longer than usual since, as I'm sure you are aware, the COVID-19 pandemic has impacted on the amount of time many researchers have for doing manuscript reviews. At this point, we have received 2 reviews of your manuscript. Both reviews are generally positive and in both cases, outline only a short list of issues that require revision. However, I would like to point out that some of these requests will take some consideration of how to address the framing of your findings (see Reviewer 2's comments). I would also like to see some of the justifications/clarifications requested by Reviewer 1 addressed.

We look forward to receiving your revised manuscript.

Kind regards,

Andrea Gruneir

Academic Editor

PLOS ONE

Journal Requirements:

2. Please include additional information regarding the survey or questionnaire used to collect demographic information in the study and ensure that you have provided sufficient details that others could replicate the analyses. For instance, if you developed a questionnaire as part of this study and it is not under a copyright more restrictive than CC-BY, please include a copy, in both the original language and English, as Supporting Information.

Reviewers' comments:

Reviewer's Responses to Questions

**Comments to the Author**

1. Is the manuscript technically sound, and do the data support the conclusions?

Reviewer #1: Yes

Reviewer #2: Yes

2. Has the statistical analysis been performed appropriately and rigorously? 

Reviewer #1: I Don't Know

Reviewer #2: Yes

3. Have the authors made all data underlying the findings in their manuscript fully available?

Reviewer #1: Yes

Reviewer #2: Yes

4. Is the manuscript presented in an intelligible fashion and written in standard English?

Reviewer #1: Yes

Reviewer #2: Yes

5. Review Comments to the Author

Reviewer #1: Abstract

1. Line 1 states “Some patients with schizophrenia are re-admitted to hospitals when their condition worsens so that they cannot live in communities”.

a. It is not clear to the reader what the authors mean by “some”, what percentage of these patients are readmitted?

b. It is not clear what the authors mean by “when their condition worsens so that they cannot live in communities’’?

Introduction

1. line 77; the authors need to provide enough and clear background/review about the in-hospital structure so as to build a case as to why this study needs to investigate that, what is the problem with the structure?

2. Line 88-91: there is no relationship between these sentences which makes the idea unclear.

3. Stating that the predictors of readmission are unknown on line 90-91 and there after reporting the predictors of early readmission line 92-98 is a bit confusing to the reader, consider revising this part

4. Line 102-104: basing on this statement line 103-104, "by focusing on both social climate in psychiatry wards and nurses' individual factors" the authors can revise their statement in line 90-91 which states that the predictors are unknown

Methods

1. Line 120: what sampling technique was used to select these participants?

2. Line 148-151: how was recall bias minimized?

5. line 190: How did the authors control for self-assessment bias?

6. Line 230, why did the authors use an unpaired t-test?: this does not make sense since they were dealing with mean scores from a single sample

7. Line 230, why did the authors use one-way analyses of variance?: this does not make sense because one-way analysis of variance is used to compare the means of three or more independent samples

Limitations

1. line 427-430: what are authors' recommendations based on these limitations?

2. lines 430-432: what are authors' recommendations based on these limitations?

Reviewer #2: Thank you for inviting my review of “Structure and predictors of in-hospital nursing care to prevent early readmission in patients with schizophrenia in Japan: A cross-sectional study.” This paper describes the development and administration of a survey to assess the extent to which respondents. The questionnaire was completed by 724 nurses and the team identified 5 factors characterizing in-hospital nursing care to prevent early readmission and 3 variables associated with decreased early readmission.

The research team developed a questionnaire, the In-hospital nursing care to Prevent Early Readmission in patients with Schizophrenia Scale (IPERSS), informed by results of a qualitative study, with 38 items corresponding to 38 concepts identified in the qualitative interviews, plus an additional 5 items added by 5 experts (possibly members of the research team but this was not specified). All items were revised for language by a separate 5-person panel. Other measures, including the Essen climate evaluation schema (EssenCES-JPN) and the Nursing Excellence Scale (NES) were selected based on relevance and good test characteristics.

The sample size, informed by sample size calculations for exploratory factor analysis, was large and the process for sampling hospitals rigorous. The authors used exploratory factor analysis to identify a 5-factor structure of 36 items. Total scores were associated with nursing excellence, therapeutic hold, advanced practice status, and participation of multidisciplinary teams and families in pre-discharge conferences.

Overall, this was a rigorous attempt to develop a questionnaire that catalogues nurses’ use of different nursing practices in patients with schizophrenia who have been readmitted to hospital. My main comment is to highlight that there is a subtle but important issue with the language in the questionnaire that, in some places in the paper, seems to have led to some errors in interpretation. The questionnaire asks nurses to “recall a patient with schizophrenia who had previously been readmitted within 90 days of discharge but could live in a community for more than 90 days after receiving an in-hospital intervention. Circle the numbers (1-5) that apply to the nursing practice you performed for the patient.” This does not explicitly state that the interventions were designed to “prevent” readmission. I think the authors need to be clear about the distinction between interventions that are delivered to patients who have been readmitted and interventions that are specifically designed to prevent readmission. This will also help inform future research and use of this tool.

Minor comments:

Introduction

Line 60: the examples listed are not all “patients’ clinical characteristics” (e.g., proportion of experienced psychiatrists at a hospital) – I suggest reclassifying as patient clinical and health system characteristics, for example.

The context in Japan sounds very different from elsewhere in the world (e.g., length of stay in hospital, strong emphasis on inpatient rather than outpatient care), and warrants this full description in the Introduction.

Methods

The paper suggests, but it is not clear, whether 1,995 represents the number of registered nurses who were deemed eligible and approached for participation. This is important information as it speaks to the representativeness of this sample.

Discussion

Line 410: “The R-squared in the stepwise multiple 410 regression analyses of this study was not high enough” needs to be elaborated. Not high enough for what? It is also helpful to explain the implications of the R-squared in this context.

Line 423: The authors state that they were not able to identify any interventions to engage inpatients with community providers. There are a number of reviews on this topic, for example, “Transitional interventions to reduce early psychiatric readmissions in adults: systematic review” (https://pubmed.ncbi.nlm.nih.gov/23457182/), which is referenced in the Introduction, and likely others that are more current.

6. PLOS authors have the option to publish the peer review history of their article (what does this mean?). If published, this will include your full peer review and any attached files.

Reviewer #1: No

Reviewer #2: No

---

## [Author Response · Author response to Decision Letter 0]

1 Mar 2021

Dear Academic Editor and Reviewers

Thank you very much for reviewing our manuscript and offering valuable advice.

We have addressed your comments with point-by-point responses and revised the manuscript accordingly.

Journal Requirements:

Response

Thank you for your comment. The manuscript has been formatted according to PLOS ONE’s style requirements. Furthermore, since the affiliation of the first author has changed, the following has been added to the title page.

(Page 8. Line 173–174. Methods)

“(see S1 Fig.)”

(Page 39. Line 719. Supporting information)

“S1 Fig.”

(Page 1. Line 18–19. Title page)

“#aCurrent Address: Department of Nursing, School of Medicine, Gifu University, Gifu-shi, Gifu, Japan”

Journal Requirements:

Please include additional information regarding the survey or questionnaire used to collect demographic information in the study and ensure that you have provided sufficient details that others could replicate the analyses. For instance, if you developed a questionnaire as part of this study and it is not under a copyright more restrictive than CC-BY, please include a copy, in both the original language and English, as Supporting Information.

Response

Thank you for your comment. We have included additional information in the main text and have added the Supporting Information section as follows:

(Page 40. Line 729–732. Supporting information)

“S3 Appendix. Questionnaire used to collect demographic information (English).

(DOCX)

S4 Appendix. Questionnaire used to collect demographic information (Japanese).

(DOCX)”

Reviewer #1s’ comment: 

Abstract

Line 1 states “Some patients with schizophrenia are re-admitted to hospitals when their condition worsens so that they cannot live in communities”.

a. It is not clear to the reader what the authors mean by “some”, what percentage of these patients are readmitted?

b. It is not clear what the authors mean by “when their condition worsens so that they cannot live in communities’’?

Response

Thank you for your valuable comments. The sentence, “Some patients with schizophrenia are re-admitted to hospitals when their condition worsens so that they cannot live in communities,’’ was unclear. We have revised it for clear as follows:

(Page 2. Line 26-28. Abstract)

“About 15%–30% of patients with schizophrenia are readmitted within 90 days of discharge due to exacerbation of symptoms that leads to self-harm, harm to others, or self-neglect.”

Reviewer #1s’ comment: 

Introduction

Line 77; the authors need to provide enough and clear background/review about the in-hospital structure so as to build a case as to why this study needs to investigate that, what is the problem with the structure?

Response

Thank you for drawing our attention to this problem. The original manuscript did not describe the background/review about the in-hospital structure. Therefore, it was unclear why this study needs to investigate the structure of in-hospital nursing care. In Lines 82–83 of the revised manuscript, we have now described the importance of in-hospital care leading to reduction in early readmission provided during hospitalization. Additionally, the following paragraph has been included to provide a background/review about the in-hospital structure, with 10 references added to the References section:

(Page 4. Line 82–91. Introduction)

“In-hospital care that leads to reduction in early readmission is important for achieving successful transitions from hospital care to community care.

Revealing the structure of such in-hospital care will facilitate the development of intervention programs that support this transition. Previous studies have identified the structure of the following six types of in-hospital psychiatric care: care and management of inpatients who exhibit self-cutting behaviors [23]; case management practice [24]; health services to promote mental health [25, 26]; psychiatric services in forensic inpatient care [27]; therapeutic attitudes of professionals working with drug abusers [28]; and rehabilitative care in longer term mental health facilities [29-32]. However, the structure of in-hospital care leading to reductions in early readmissions is unclear.”

(Page 34. Line 578–Page 35. Line 618. Reference)

“23. Hosie L, Dickens GL. Harm-reduction approaches for self-cutting in inpatient mental health settings: Development and preliminary validation of the Attitudes to Self-cutting Management (ASc-Me) Scale. Journal of Psychiatric and Mental Health Nursing. 2018;25(9-10):531-545. doi: 10.1111/jpm.12498.

24. Chen S-C, Lee S-K, Rong J-R, Wu C-C, Liu W-I. The Development and Psychometric Testing on Psychiatric Nurses of a Nurse Case Management Competence Scale in Taiwan. The journal of nursing research. 2018;26(2):72-79. doi: 10.1097/jnr.0000000000000230.

25. Lundqvist LO, Suryani, Hermiati D, Sutini T, Schroder A. A psychometric evaluation of the Indonesian version of the Quality in Psychiatric Care-Inpatient Staff (QPC-IPS) instrument. Asian Journal of Psychiatry. 2019;46:29-33. doi: 10.1016/j.ajp.2019.09.027.

26. Lundqvist LO, Suryani, Anna N, Rafiyah I, Schroder A. Indonesian adaptation of the Quality in Psychiatric Care-Inpatient (QPC-IP) instrument: Psychometric properties and factor structure. Asian Journal of Psychiatric. 2018;34:1-5. doi: 10.1016/j.ajp.2018.03.006.

27. Lundqvist L-O, Riiskjaer E, Lorentzen K, Schröder A. Factor Structure and Psychometric Properties of the Danish Adaptation of the Instrument Quality in Psychiatric Care-Forensic In-Patient Staff (QPC-FIPS). Open Journal of Nursing. 2014;04(12):878-885. doi: 10.4236/ojn.2014.412093.

28. Takano A, Kawakami N, Miyamoto Y, Matsumoto T. A Study of Therapeutic Attitudes Towards Working With Drug Abusers: Reliability and Validity of the Japanese Version of the Drug and Drug Problems Perception Questionnaire. Archives of Psychiatric Nursing. 2015;29(5):302-308. doi: 10.1016/j.apnu.2015.05.002.

29. Killaspy H, White S, Dowling S, Krotofil J, McPherson P, Sandhu S, et al. Adaptation of the Quality Indicator for Rehabilitative Care (QuIRC) for use in mental health supported accommodation services (QuIRC-SA). BMC Psychiatry. 2016;16:101. doi: 10.1186/s12888-016-0799-4.

30. Killaspy H, White S, Wright C, Taylor TL, Turton P, Kallert T, et al. Quality of longer term mental health facilities in Europe: validation of the quality indicator for rehabilitative care against service users' views. PLOS ONE. 2012;7(6):e38070. doi: 10.1371/journal.pone.0038070.

31. Killaspy H, Cardoso G, White S, Wright C, Caldas de Almeida JM, Turton P, et al. Quality of care and its determinants in longer term mental health facilities across Europe; a cross-sectional analysis. BMC Psychiatry. 2016;16:31. doi: 10.1186/s12888-016-0737-5.

32. Killaspy H, White S, Wright C, Taylor TL, Turton P, Schutzwohl M, et al. The development of the Quality Indicator for Rehabilitative Care (QuIRC): a measure of best practice for facilities for people with longer term mental health problems. BMC Psychiatry. 2011;11:35. doi: 10.1186/1471-244X-11-35.”

Reviewer #1s’ comment:

Line 88-91: there is no relationship between these sentences which makes the idea unclear.

Response

Thank you for bringing this point to our attention. As the reviewer states, these sentences seemed unrelated. Therefore, we have deleted these sentences (Page 4, Line 88–91. in the original manuscript).

Reviewer #1s’ comment:

Stating that the predictors of readmission are unknown on line 90-91 and there after reporting the predictors of early readmission line 92-98 is a bit confusing to the reader, consider revising this part.

Response

Thank you for pointing out this error. We agree that this part was confusing. Reviewer #2 suggested that we should state that the context of mental health care in Japan is different from elsewhere. In accordance with the suggestion of Reviewer # 2, we have revised this point as follows:

(Page 5. Line 101–103. Introduction)

“The context of mental health care in Japan is different from elsewhere in the world in that the length of hospitalization is longer and the more resources are invested in inpatient care rather than community care.”

Reviewer #1s’ comment:

Line 102-104: basing on this statement line 103-104, "by focusing on both social climate in psychiatry wards and nurses' individual factors" the authors can revise their statement in line 90-91 which states that the predictors are unknown.

Response

We appreciate your important advice. We should have stated why the predictors of in-hospital nursing care need to be clarified. The paragraph following Line 91 in the original manuscript explains that it is hypothesized that in-hospital nursing care is predicted by the social climate in psychiatric wards and nurses’ individual factors. We have added the following sentence at the beginning of the next paragraph:

(Page 5. Line 104–107. Introduction)

“Therefore, it is desirable to identify the predictors of in-hospital nursing care leading to reduction in early readmission in the Japanese mental health care context. Specifically, this information could be used to develop strategies to support the successful transition of patients from hospital care to community care.”

Reviewer #1s’ comment:

Methods

Line 120: what sampling technique was used to select these participants?

Response

Thank you for your important comment. In the original manuscript, we had not described how the participants were selected. We have stated this process in the revised manuscript, as follows:

(Page 6. Line 137–141. Methods)

“Nurse managers at the 40 selected hospitals provided information on the number of registered nurses (RNs) working at their hospital who met the inclusion criteria listed below. They reported that 1,995 RNs fulfilled the criteria. Subsequently, request documents for research cooperation to the RNs with self-administered questionnaires were mailed to the nurse managers, who then distributed them to the 1,995 RNs identified.”

Reviewer #1s’ comment:

Line 148-151: how was recall bias minimized?

Response

Thank you for your comment. In the original manuscript, we did not explain how we minimized recall bias. This was done by providing a diagram with the questionnaire to present the time axis. Additionally, the questionnaire items were made more specific. However, it was difficult to completely avoid recall bias when answering the questionnaire developed in this study. Accordingly, we have described the following in the Methods and Limitations sections:

(Page 8. Line 172–174. Methods)

“To minimize recall bias, a diagram representing the time axis was provided with the questionnaire (see S1 Fig.), and the items are expressed in concrete contents.”

(Page 29. Line 478–482. Limitations)

“The design adopted in this study could not completely eliminate the recall bias. Respondents should be required to recall more recent nursing practices, such as those within the last year. Alternatively, the questionnaire could utilize a vignette case of a fictitious patient with schizophrenia who was readmitted early.”

Reviewer #1s’ comment:

Line 190: How did the authors control for self-assessment bias?

Response

Thank you for bringing this point to our attention. We agree that our original manuscript did not describe how we controlled for self-assessment bias. We have added the following in the Methods section:

(Page 10. Line 226–228. Methods)

“To control for self-assessment bias, the questionnaire instructions explained that the content of the responses to this survey would not affect the evaluation of the participants’ performance.”

Reviewer #1s’ comment:

Line 230, why did the authors use an unpaired t-test?: this does not make sense since they were dealing with mean scores from a single sample

Response

We apologize for our unclear description of the unpaired t-tests. We performed unpaired t-tests to identify variables that need to be input in the multiple regression analyses (Page 12, Line 269–273. in the revised manuscript) and to examine the data obtained. We have described the following in the Methods section:

(Page 11. Line 256–257. Methods)

“Correlation and univariate analyses (i.e., unpaired t-tests, and one-way analyses of variance with post hoc analyses) were performed to examine the data obtained.”

Additionally, we had not explained how the unpaired t-tests were performed. We divided the participants into two groups based on each of the following variables: gender, APRN status, experience as a psychiatric home visiting nurse, experience in providing psychiatric outpatient care, experience in somatic care wards, and adoption of primary nursing. We compared the differences in the overall scale and each subscale of the IRERSS between the two groups. In the original manuscript, we had not clearly explained that these unpaired t-tests were performed after dividing the sample into two groups. We have now described the following in the Methods section:

(Page 11. Line 259.–Page 12. Line 263. Methods)

“Using unpaired t-tests, differences in the overall scale and each subscale of the IRERSS were compared between groups dichotomized based on each of the following variables: gender, APRN status, experience as a psychiatric home visiting nurse, experience in providing psychiatric outpatient care, experience in somatic care wards, and adoption of primary nursing.”

Reviewer #1s’ comment:

Line 230, why did the authors use one-way analyses of variance?: this does not make sense because one-way analysis of variance is used to compare the means of three or more independent samples

Response

We apologize for our insufficient explanation of the one-way analyses of variance. As with the unpaired t-tests, we performed one-way analyses of variance to identify variables that could be input in the multiple regression analyses (Page 12, Line 269–273. in the revised manuscript) and to examine the data obtained (Page 11, Line 256–257 in the revised manuscript). 

Additionally, we had not explained how we performed the one-way analyses of variance and post hoc analyses (Bonferroni correction). We divided the participants into three groups based on each of the following variables: years of psychiatric experience (i.e., less than 5 years, 5–14 years, 15 years or more), education level, hospital establishment, participation of families in pre-discharge conferences, and participation of multidisciplinary teams in pre-discharge conferences. We compared the differences in the overall scale and each subscale of the IRERSS among the three groups. In the original manuscript, we had not clearly described that the one-way analyses of variance and post hoc analyses (Bonferroni correction) were performed after dividing the sample into three groups. We have added the following in the Methods section:

(Page 10. Line 234–235. Methods)

“Participants’ years of psychiatric experience were classified into three categories: less than 5 years, 5–14 years, 15 years or more.”

(Page 12. Line 263–268. Methods)

“Using one-way analyses of variance and post hoc analyses (Bonferroni correction), differences in the overall scale and each subscale of the IRERSS were compared among the three groups classified using each of the following variables: years of psychiatric experience, educational level, hospital establishment, participation of families in pre-discharge conferences, and participation of multidisciplinary teams in pre-discharge conferences.”

Reviewer #1s’ comment:

Limitations

Line 427-430: what are authors' recommendations based on these limitations?

Response

Thank you for your comment. Our original manuscript did not clearly describe our recommendations based on the first limitation. We have added the following in the Limitations section:

(Page 29. Line 474–476. Limitations)

“The low response rate may be attributed to the high number of items in the questionnaire and their complicated wording. This suggests the need to refine the items of the IRERSS to create more straightforward sentences.”

Reviewer #1s’ comment:

Lines 430-432: what are authors' recommendations based on these limitations?

Response

Thank you for your kind suggestion. We agree that our original manuscript did not state our recommendations based on the second limitation. Overlapping with the response to the comments by Reviewer #1 on Line 148–151 in the original manuscript, we have added the following in the Limitations section: 

(Page 29. Line 478–482. Limitations)

“The design adopted in this study could not completely eliminate the recall bias. Respondents should be required to recall more recent nursing practices, such as those within the last year. Alternatively, the questionnaire could utilize a vignette case of a fictitious patient with schizophrenia who was readmitted early.”

Reviewer #2s’ comment: 

My main comment is to highlight that there is a subtle but important issue with the language in the questionnaire that, in some places in the paper, seems to have led to some errors in interpretation. The questionnaire asks nurses to “recall a patient with schizophrenia who had previously been readmitted within 90 days of discharge but could live in a community for more than 90 days after receiving an in-hospital intervention. Circle the numbers (1-5) that apply to the nursing practice you performed for the patient.” This does not explicitly state that the interventions were designed to “prevent” readmission. I think the authors need to be clear about the distinction between interventions that are delivered to patients who have been readmitted and interventions that are specifically designed to prevent readmission. This will also help inform future research and use of this tool.

Response

Thank you very much for your excellent suggestion. Reviewer #2’s comment showed that it is important to distinguish clearly between interventions that are delivered to patients who have been readmitted and those which are specifically designed to prevent readmission. The questionnaire did not explicitly state that the interventions were designed to “prevent” readmission. We agree that there are many merits in distinguishing clearly between such interventions; however, the design adopted in this study could not distinguish clearly between them. The items of the questionnaire would include interventions that are effective in preventing early readmission and those provided usually during hospitalization. The latter may also be important as the basic nursing care needed to establish therapeutic relationships with patients who have been readmitted.

Interventions included in the questionnaire can be considered as in-hospital nursing care “leading to reduction in early readmission” because the questionnaire items were based on a qualitative study that clarified the contents of in-hospital nursing care leading to reduction in early readmission (Page 8, Line 183.–Page 9, Line 194. in the revised manuscript). Thus, the present questionnaire assessed the degree of implementation of in-hospital nursing care “leading to reduction in early readmission” among patients with schizophrenia. Accordingly, we have revised the following five points.

1. We have altered all expressions of “in-hospital nursing care to prevent early readmission in patients with schizophrenia” to “in-hospital nursing care leading to reduction in early readmission among patients with schizophrenia” throughout the manuscript.

(Page 1. Line 4–6. Title page)

(Page 2. Line 29–30, 32–33. Abstract)

(Page 5. Line 117–118. Introduction)

(Page 7. Line 166–Page 8. Line 167., Page 8. Line 184–189. Methods)

(Page 25. Line 379–380., Page 28. Line 458–459. Discussion)

(Page 30. Line 498–500. Conclusion)

“in-hospital nursing care leading to reduction in early readmission among patients with schizophrenia”

(Page 2. Line 36–40, 43, 47–48. Abstract)

(Page 5. Line 107–108, 114–115, 119–120. Introduction)

(Page 9. Line 213–214. Methods)

(Page 25. Line 384–385, Page 26. Line 408., Page 28. Line 448–449., Page 30. Line 485. Discussion)

(Page 30. Line 503. Conclusion)

(Page 39. Line 719. Supporting information)

“in-hospital nursing care leading to reduction in early readmission”

2. We have changed the abbreviations “IPERSS = In-hospital nursing care to prevent early readmission in patients with schizophrenia scale” to “IRERSS = In-hospital nursing care leading to reduction in early readmission among patients with schizophrenia scale” throughout the manuscript.

(Page 18. Line 312–313, Page 19. Line 328–329, Page 22. Line 344–345, Page 25. Line 367–368. Results)

“Abbreviations: IRERSS = In-hospital nursing care leading to reduction in early readmission among patients with schizophrenia scale”

(Page 8. Line 167–168, 183., Page 9. Line 194., Page 10. Line 225, 229., Page 11. Line 238–240, 253, 255, 258, 260., Page 12. Line 265, 271. Methods)

(Page 14. Line 293–294, 297., Page 15. Line 300, 303, 309–310, and Table 2., Page 18. Line 315, 319, 321–324, and Table 3., Page 19. Line 336–337, and Table 4., Page 23. Line 353, 361, and Table 5. Results)

(Page 25. Line 374–375, 381–382, 384., Page 26., Line 387, 390–391, 404, 406, 409–410., Page 27. Line 414, 417–418, 422, 431., Page 28. Line 437., Page 29. Line 464. Discussion)

(Page 39. Line 721. Page 40. Line 723, 725. Supporting information)

“IRERSS”

3. To avoid using the phrases “the prevention of early readmissions” or “to prevent early readmission,” we have replaced them with other words or sentences in the Introduction and Discussion sections.

(Page 4. Line 80–81. Introduction)

“Reduction in early readmissions”

(Page 5. Line 100. Introduction)

“to decrease early readmission”

(Page 26. Line 388–390. Discussion)

“The previous study revealed qualitatively that hospital nurses’ practices leading to reduction in early readmission among patients with schizophrenia could be classified into five categories.”

(Page 26. Line 401–403. Discussion)

“This study showed that Japanese in-hospital nursing care leading to reduction in early readmission among patients with schizophrenia comprised these five factors.”

4. We have added information on how to interpret the statement, “nursing care provided for a patient with schizophrenia who had previously been readmitted within 90 days of discharge but could live in a community for more than 90 days after receiving an in-hospital intervention” in the Methods section.

(Page 8. Line 178–181. Methods)

“The nursing care delivered during the above in-hospital intervention would be considered “in-hospital nursing care leading to reductions in early readmission” if patient was not re-hospitalized within 90 days following discharge after receiving the nursing care.”

(Page 8. Line 184–186. Methods)

In their qualitative study, they examined “in-hospital nursing care leading to reduction in early readmission” among patients by conducting interviews with 17 proficient psychiatric nurses.

(Page 9. Line 194–196. Methods)

“Accordingly, the IRERSS assesses the degree of implementation of “in-hospital nursing care leading to reduction in early readmission among patients with schizophrenia.””

5. In the Limitations section, we have added that this study could not clarify the difference between interventions that are delivered to patients who have been readmitted early and those which are specifically designed to prevent early readmission.

(Page 30. Line 486–495. Limitations)

“Fourth, this study could not clarify the difference between interventions delivered to patients who had been readmitted early and those specifically designed to prevent early readmission. In other words, the design adopted in this study could not reveal which factors or items of the IRERSS were effective in preventing early readmission of patients with schizophrenia. The IRERSS would include the interventions that are effective in preventing early readmission and those provided usually during hospitalization. Early readmission might also be affected by patients’ medical state, therapies they received, and community care after discharge. Therefore, future studies using the IRERSS need to compare nursing practices implemented for patients with schizophrenia who were readmitted early and for those who were not, adjusting for the effect of patients’ medical state, therapies they received, and community care after discharge.” 

Reviewer #2s’ comment: 

Introduction

Line 60: the examples listed are not all “patients’ clinical characteristics” (e.g., proportion of experienced psychiatrists at a hospital) – I suggest reclassifying as patient clinical and health system characteristics, for example.

Response

Thank you for your kind suggestion. We agree that we should reclassify the reference list. We have reclassified the references based on three perspectives: patients’ clinical characteristics, health system characteristics, and characteristics of hospitalizations. We have revised the corresponding section as follows: 

(Page 3. Line 62–72. Introduction)

“Factors associated with readmission can be classified based on three criteria: patients’ clinical characteristics, health system characteristics, and characteristics of hospitalizations. Regarding patients’ clinical characteristics, younger age [10, 11], marital status of “unmarried” [11], complications [6, 12], medication nonadherence [13-15], and maladaptive functioning of family systems [16] are associated with increased risk for readmission. Regarding health system characteristics, proportion of experienced psychiatrists at a hospital [17], multiple uses of health service [6], and unplanned discharge [12] are associated with increased risk for readmission. Regarding characteristics of hospitalizations, having a history of previous hospitalizations [6, 11, 12], duration of involuntary admission [11, 12], and total admission duration [6, 11] are associated with increased risk for readmission.”

Reviewer #2s’ comment: 

The context in Japan sounds very different from elsewhere in the world (e.g., length of stay in hospital, strong emphasis on inpatient rather than outpatient care), and warrants this full description in the Introduction.

Response

Thank you for your recommendation. We agree that the description of the context of mental health care in Japan was imperative. We have added the following in the Introduction section: 

(Page 5. Line 101–103. Introduction)

“The context of mental health care in Japan is different from elsewhere in the world in that the length of hospitalization is longer and the more resources are invested in inpatient care rather than community care.”

Reviewer #2s’ comment:

Methods

The paper suggests, but it is not clear, whether 1,995 represents the number of registered nurses who were deemed eligible and approached for participation. This is important information as it speaks to the representativeness of this sample.

Response

We apologize for our unclear description of the number of participants. We have also received comments from Reviewer #1 about the sampling technique. We have stated clearly what kind of sampling technique we used in this study and have described that the number 1,995 represented eligible participants, as follows: 

(Page 6. Line 137–141. Methods)

“Nurse managers at the 40 selected hospitals provided information on the number of registered nurses (RNs) working at their hospital who met the inclusion criteria listed below. They reported that 1,995 RNs fulfilled the criteria. Subsequently, request documents for research cooperation to the RNs with self-administered questionnaires were mailed to the nurse managers, who then distributed them to the 1,995 RNs identified.”

(Page 7. Line 150. Methods)

“The sample of 1,995 utilized in this study can be considered adequate.”

Reviewer #2s’ comment:

Discussion

Line 410: “The R-squared in the stepwise multiple 410 regression analyses of this study was not high enough” needs to be elaborated. Not high enough for what? It is also helpful to explain the implications of the R-squared in this context.

Response

Thank you for your kind recommendation. We agree that the description of the R-squared in the stepwise multiple regression analyses should be more detailed. We have deleted the sentence “However, no data related to patients’ characteristics were obtained in this study” (Page 26, Line 413–414 in the original manuscript), and have added the following in the Discussion and Reference sections: 

(Page 28. Line 450–456. Discussion)

“The R-squared in the stepwise multiple regression analyses conducted in this study were not high enough. The R-squared represents the amount of variance in the dependent variable that is explained by the independent variables. It ranges from 1.0 (perfect prediction) to 0.0 (no prediction) [59]. In this study, data on variables related to patients’ characteristics were not collected and they were not entered in the stepwise regression analyses. This may explain why the R-squared was not high enough.”

(Page 39. Line 706–707. Reference)

“59. Joseph F. Hair J, Black WC, Babin BJ, Anderson RE. Multivariate Data Analysis: Cengage Learing, EMEA, United Kingdom; 2019.”

Reviewer #2s’ comment:

Line 423: The authors state that they were not able to identify any interventions to engage inpatients with community providers. There are a number of reviews on this topic, for example, “Transitional interventions to reduce early psychiatric readmissions in adults: systematic review” (https://pubmed.ncbi.nlm.nih.gov/23457182/), which is referenced in the Introduction, and likely others that are more current.

Response

Thank you very much for your important comments. We apologize for the errors made in the Discussion section. We identified that many reviews noted the importance of interventions that engage inpatients with community providers. We have revised this point as follows, with three references added to the References section:

(Page 29. Line 466–469. Discussion)

“In previous studies, interventions to engage patients with community care providers were found to reduce readmissions [10, 60-62]. This study quantitatively confirmed that communication between patients and community care providers promotes in-hospital nursing care leading to reduction in early readmission among patients with schizophrenia.”

(Page 39. Line 708–717. Reference)

“60. Tyler N, Wright N, Waring J. Interventions to improve discharge from acute adult mental health inpatient care to the community: systematic review and narrative synthesis. BMC Health Services Research. 2019;19(1):883. doi: 10.1186/s12913-019-4658-0.

61. Shaffer SL, Hutchison SL, Ayers AM, Goldberg RW, Herman D, Duch DA, et al. Brief Critical Time Intervention to Reduce Psychiatric Rehospitalization. Psychiatric Services. 2015;66(11):1155-1161. doi: 10.1176/appi.ps.201400362.

62. Chen FP. Developing community support for homeless people with mental illness in transition. Community Mental Health Journal. 2014;50(5):520-30. doi: 10.1007/s10597-013-9641-3.”

---

## [Decision Letter · Decision Letter 1]

7 Apr 2021

PONE-D-20-29533R1

Structure and predictors of in-hospital nursing care leading to reduction in early readmission among patients with schizophrenia in Japan: A cross-sectional study

PLOS ONE

Dear Dr. Maki,

Thank you for submitting your manuscript to PLOS ONE. After careful consideration, we feel that it has merit but does not fully meet PLOS ONE’s publication criteria as it currently stands. Therefore, we invite you to submit a revised version of the manuscript that addresses the points raised during the review process.

ACADEMIC EDITOR:

Please address the 2 very minor points raised by the second reviewer. 

We look forward to receiving your revised manuscript.

Kind regards,

Andrea Gruneir

Academic Editor

PLOS ONE

Journal Requirements:

Reviewers' comments:

Reviewer's Responses to Questions

**Comments to the Author**

1. If the authors have adequately addressed your comments raised in a previous round of review and you feel that this manuscript is now acceptable for publication, you may indicate that here to bypass the “Comments to the Author” section, enter your conflict of interest statement in the “Confidential to Editor” section, and submit your "Accept" recommendation.

Reviewer #1: All comments have been addressed

Reviewer #2: All comments have been addressed

2. Is the manuscript technically sound, and do the data support the conclusions?

Reviewer #1: (No Response)

Reviewer #2: Yes

3. Has the statistical analysis been performed appropriately and rigorously? 

Reviewer #1: (No Response)

Reviewer #2: Yes

4. Have the authors made all data underlying the findings in their manuscript fully available?

Reviewer #1: (No Response)

Reviewer #2: Yes

5. Is the manuscript presented in an intelligible fashion and written in standard English?

Reviewer #1: (No Response)

Reviewer #2: Yes

6. Review Comments to the Author

Reviewer #1: (No Response)

Reviewer #2: A couple very minor points:

Abstract:

“About 15%–30% of patients with schizophrenia are readmitted within 90 days of discharge due to exacerbation of symptoms that leads to self-harm, harm to others, or self-neglect.” Is this data specific to Japan or from global research? Suggest specifying that this is referring to inpatient psychiatric admissions, e.g., “Globally, around 15-30% of patients with schizophrenia discharged from inpatient psychiatric admissions are readmitted within 90 days….”

Discussion:

Line 410: “The R-squared in the stepwise multiple regression analyses of this study was not high enough” still suggests that there is a specific threshold for R-squared the authors were trying to reach. Suggest rewriting as “The R-squared in the stepwise multiple regression analyses of this study was not high” or specify a threshold.

7. PLOS authors have the option to publish the peer review history of their article (what does this mean?). If published, this will include your full peer review and any attached files.

Reviewer #1: No

Reviewer #2: No

---

## [Author Response · Author response to Decision Letter 1]

9 Apr 2021

Dear Academic Editor and Reviewers

Thank you very much for reviewing our manuscript and offering valuable suggestions.

We have addressed your comments with point-by-point responses and revised the manuscript accordingly.

Journal Requirements:

Response

Thank you for your comment. We have revised the Reference section. We have replaced the parts that should not be uppercase with lowercase and the parts that should not be lowercase with uppercase (Page 31, Line 512–Page 39, Line 719. in the revised manuscript).

We have cited no papers that have been retracted.

Reviewer #2s’ comment:

Abstract:

“About 15%–30% of patients with schizophrenia are readmitted within 90 days of discharge due to exacerbation of symptoms that leads to self-harm, harm to others, or self-neglect.” Is this data specific to Japan or from global research? Suggest specifying that this is referring to inpatient psychiatric admissions, e.g., “Globally, around 15-30% of patients with schizophrenia discharged from inpatient psychiatric admissions are readmitted within 90 days….”.

Response

Thank you very much for your valuable suggestion. As the reviewer states, we should specify that this data is from global research and that this sentence refers to inpatient psychiatric admissions. Therefore, we have revised this point as follows:

(Page 2. Line 26–28. Abstract)

“Globally, about 15%–30% of patients with schizophrenia discharged from inpatient psychiatric admissions are readmitted within 90 days due to exacerbation of symptoms that leads to self-harm, harm to others, or self-neglect.”

Reviewer #2s’ comment:

Discussion:

Line 410: “The R-squared in the stepwise multiple regression analyses of this study was not high enough” still suggests that there is a specific threshold for R-squared the authors were trying to reach. Suggest rewriting as “The R-squared in the stepwise multiple regression analyses of this study was not high” or specify a threshold.

Response

Thank you for your comment. As the reviewer states, “The R-squared in the stepwise multiple regression analyses of this study was not high enough” still suggests that there is a specific threshold for R-squared we were trying to reach. Therefore, we have deleted the word “enough” (Page 28, Line 451, and Line 456. in the original manuscript).

---

## [Editor Report · Decision Letter 2]

14 Apr 2021

Structure and predictors of in-hospital nursing care leading to reduction in early readmission among patients with schizophrenia in Japan: A cross-sectional study

PONE-D-20-29533R2

Dear Dr. Maki,

We’re pleased to inform you that your manuscript has been judged scientifically suitable for publication and will be formally accepted for publication once it meets all outstanding technical requirements.

Kind regards,

Andrea Gruneir

Academic Editor

PLOS ONE
---

## [Editor Report · Acceptance letter]

22 Apr 2021

PONE-D-20-29533R2 

Structure and predictors of in-hospital nursing care leading to reduction in early readmission among patients with schizophrenia in Japan: A cross-sectional study 

Dear Dr. Maki:

I'm pleased to inform you that your manuscript has been deemed suitable for publication in PLOS ONE. Congratulations! Your manuscript is now with our production department. 

Kind regards, 

on behalf of

Dr. Andrea Gruneir 

Academic Editor

PLOS ONE